

# P2X7 antagonism using Brilliant Blue G reduces body weight loss and prolongs survival in female SOD1$^{G93A}$ amyotrophic lateral sclerosis mice

Rachael Bartlett, Vanessa Sluyter, Debbie Watson, Ronald Sluyter and Justin J. Yerbury

Illawarra Health and Medical Research Institute, Wollongong, NSW, Australia
School of Biological Sciences, University of Wollongong, Wollongong, NSW, Australia
Centre for Medical and Molecular Biosciences, Wollongong, NSW, Australia

Corresponding authors
Ronald Sluyter, rsluyter@uow.edu.au
Justin J. Yerbury,
jyerbury@uow.edu.au

## ABSTRACT

**Background**. Amyotrophic lateral sclerosis (ALS) is a rapidly progressive neurodegenerative disease characterised by the accumulation of aggregated proteins, microglia activation and motor neuron loss. The mechanisms underlying neurodegeneration and disease progression in ALS are unknown, but the ATP-gated P2X7 receptor channel is implicated in this disease. Therefore, the current study aimed to examine P2X7 in the context of neurodegeneration, and investigate whether the P2X7 antagonist, Brilliant Blue G (BBG), could alter disease progression in a murine model of ALS.

**Methods**. Human SOD1$^{G93A}$ transgenic mice, which normally develop ALS, were injected with BBG or saline, three times per week, from pre-onset of clinical disease (62–64 days of age) until end-stage. During the course of treatment mice were assessed for weight, clinical score and survival, and motor coordination, which was assessed by rotarod performance. Various parameters from end-stage mice were assessed as follows. Motor neuron loss and microgliosis were assessed by immunohistochemistry. Relative amounts of lumbar spinal cord SOD1 and P2X7 were quantified by immunoblotting. Serum monocyte chemoattractant protein-1 was measured by ELISA. Splenic leukocyte populations were assessed by flow cytometry. Relative expression of splenic and hepatic P2X7 mRNA was measured by quantitative real-time PCR. Lumbar spinal cord SOD1 and P2X7 were also quantified by immunoblotting in untreated female SOD1$^{G93A}$ mice during the course of disease.

**Results**. BBG treatment reduced body weight loss in SOD1$^{G93A}$ mice of combined sex, but had no effect on clinical score, survival or motor coordination. BBG treatment reduced body weight loss in female, but not male, SOD1$^{G93A}$ mice. BBG treatment also prolonged survival in female, but not male, SOD1$^{G93A}$ mice, extending the mean survival time by 4.3% in female mice compared to female mice treated with saline. BBG treatment had no effect on clinical score or motor coordination in either sex. BBG treatment had no major effect on any end-stage parameters. Total amounts of lumbar spinal cord SOD1 and P2X7 in untreated female SOD1$^{G93A}$ mice did not change over time.

**Discussion**. Collectively, this data suggests P2X7 may have a partial role in ALS progression in mice, but additional research is required to fully elucidate the contribution of this receptor in this disease.

## INTRODUCTION

Amyotrophic lateral sclerosis (ALS) is the most common form of adult-onset motor neuron disease, and is characterised by the degeneration and death of both upper and lower motor neurons (*Boillee, Vande Velde & Cleveland, 2006*). This leads to progressive spasticity, muscle weakness and muscle atrophy, culminating in paralysis and death, generally within 3–5 years from clinical onset (*Al-Chalabi et al., 2012*). While most cases of ALS are sporadic (sporadic ALS, or sALS) and of unclear cause, the 5–10% of cases in which the disease is inherited (familial ALS, or fALS) can be linked to specific genetic mutations. Mutations in one or more of at least a dozen genes give rise to distinct disease neuropathological subtypes, each of which is associated with the aggregation of a variety of proteins into inclusions with histologically distinct structures. The inclusions in ALS can also be categorized on the basis of the dominant protein involved, such as Cu, Zn superoxide dismutase 1 (SOD1), and the RNA-binding proteins, TDP-43 and FUS, which result in inclusions with varied characteristics (*Farrawell et al., 2015*). Protein aggregates associated with ALS pathology are thought to drive progression of disease, possibly through a prion-like action (*Grad et al., 2014*) or by chronically activating microglia (*Roberts et al., 2013*).

SOD1 was the first gene in which mutations were found to be associated with ALS, and account for approximately 20% of familial cases (2% of all cases) (*Rosen et al., 1993*). To investigate mechanisms of disease and potential therapeutic strategies, several transgenic superoxide dismutase 1 (SOD1) ALS mouse models have been generated (*Turner & Talbot, 2008*), including the SOD1$^{G93A}$ mouse model (*Gurney et al., 1994*). SOD1$^{G93A}$ mice carry a high copy number of a transgene encoding for the G93A variant of human SOD1, and develop an ALS-like disease which is characterised by the loss of motor neurons leading to hind limb weakness and paralysis (*Gurney et al., 1994*). These mice display similar pathological hallmarks to those seen in the human disease (*Cheroni et al., 2005*; *Hall, Oostveen & Gurney, 1998*; *Kawaguchi-Niida et al., 2013*), and display a similar gender incidence and prevalence of disease to human ALS (*Cervetto et al., 2013*; *McCombe & Henderson, 2010*; *Veldink et al., 2003*).

Neuroinflammation is emerging as a central component of ALS progression (*Henkel et al., 2009*). In the central nervous system (CNS), activation of the ATP-gated P2X7 receptor channel is responsible for mediating a number of neuroinflammatory events, including microglial activation and proliferation (*Monif et al., 2009*). P2X7 immunoreactivity is increased in activated microglia of spinal cords from (post-mortem) humans with ALS and advanced-stage transgenic SOD1$^{G93A}$ rats (*Casanovas et al., 2008*; *Yiangou et al., 2006*). Furthermore, P2X7 is up-regulated in primary microglia from SOD1$^{G93A}$ mice (*D'Ambrosi et al., 2009*). These cells produce increased amounts of pro-inflammatory factors and undergo oxidative stress following ATP stimulation (*Apolloni et al., 2013b*; *D'Ambrosi et al., 2009*; *Parisi et al., 2013*; *Parisi et al., 2016*). These exacerbated pro-inflammatory responses lead

to the death of neuronal cell lines, and are abrogated by pharmacological blockade or gene deletion of microglial P2X7. Similarly, ATP-induced activation of P2X7 causes neurotoxic phenotypes in wild type (WT) microglia and WT or SOD1[G93A] astrocytes, leading to the death of co-cultured neurons (*Gandelman et al., 2010*; *Skaper et al., 2006*).

Despite the deleterious effects reported for P2X7 in an ALS context *in vitro*, current *in vivo* evidence suggests a more complex dual role for this receptor. In SOD1[G93A] mice lacking the *P2RX7* gene, females, but not males, had an extended lifespan compared to SOD1[G93A] mice encoding P2X7 (*Apolloni et al., 2013a*). However, these double transgenic mice also had anticipated onset, accelerated disease progression and increased astrocytosis, microgliosis and motor neuron loss compared to mice carrying the SOD1[G93A] transgene alone. This suggests that constitutive deletion of P2X7 is detrimental in ALS, but that this receptor may be neuroprotective during the early stages of the disease. Supporting this hypothesis, administration of the P2X7 antagonist, Brilliant Blue G (BBG), beginning at late pre-onset of clinical disease delayed disease onset, improved condition and motor coordination, reduced microgliosis and inflammatory markers and enhanced motor neuron survival (*Apolloni et al., 2014*). Similarly, administration of BBG beginning at disease onset improved motor coordination and slowed weight loss in male, but not female, SOD1[G93A] mice (*Cervetto et al., 2013*). However, BBG inhibition of P2X7 did not correspond with an increase in survival in either study (*Apolloni et al., 2014*; *Cervetto et al., 2013*).

To further investigate P2X7 in ALS progression, this study aimed to investigate whether administration of BBG could alter disease progression in SOD1[G93A] mice, beginning at the pre-onset of clinical disease. Body weight, ALS score and motor coordination were monitored throughout. Furthermore, a number of pathological hallmarks of disease were analysed at end-stage, including motor neuron counts, microgliosis, lumbar P2X7 and SOD1 protein amounts and serum monocyte chemoattractant protein-1 (MCP-1) concentrations.

## MATERIALS AND METHODS

### Reagents and antibodies

BBG, ethidium bromide, ATP, paraformaldehyde (PFA), RNAlater and glycerol gelatin were from Sigma-Aldrich (St. Louis, MO, USA). Sterile 0.9% NaCl was from Fresenius Kabi (Bad Homburg, Germany). Agarose was from Bioline (Alexandria, Australia). Ammonium-chloride-potassium (ACK) lysing buffer, N-PER[TM] neuronal protein extraction reagent, $100 \times$ Halt[TM] protease inhibitor single-use cocktail, normal horse serum (NHS), DNase/RNase-free distilled water and SuperSignal West Pico Chemiluminescent Substrate were from ThermoFisher Scientific (Waltham, MA, USA). Foetal bovine serum (FBS) was from Lonza (Basel, Switzerland). Tissue-Tek® optimal cutting temperature (OCT) compound was from Sakura (Flemingweg, Netherlands) and 22 mm glass coverslips were from Menzel Glaser (Braunschweig, Germany). Diploma full-cream milk powder was from Fonterra (Mount Waverley, Australia). Bovine serum albumin (BSA) and all other reagent grade chemicals and salts were from Amresco (Solon, OH, USA). The antibodies used are listed in Table 1.

**Table 1** Antibodies used to investigate the efficacy of the P2X7 antagonist Brilliant Blue G on ALS progression.

| Antibody target[a] | Conjugate[a] | Host species | Company[b] (catalogue number) |
|---|---|---|---|
| Mouse CD45 | FITC | Rat | BioLegend (103107) |
| IgG2b isotype control | FITC | Rat | AbD (MCA1125FT) |
| Mouse CD8$\alpha$ | PE | Rat | BioLegend (100707) |
| Mouse CD25 | PE | Rat | BioLegend (101903) |
| Mouse/human CD11b | PE | Rat | BioLegend (101207) |
| IgG2a isotype control | PE | Rat | eBioscience (12-4321-80) |
| Mouse CD3$\varepsilon$ | PerCP/Cy5.5 | Armenian hamster | BioLegend (100327) |
| Mouse CD11c | PerCP/Cy5.5 | Armenian hamster | BioLegend (117327) |
| IgG isotype control | PerCP/Cy5.5 | Armenian hamster | BioLegend (400931) |
| Mouse CD19 | APC | Rat | BioLegend (115511) |
| Mouse CD4 | APC | Rat | BioLegend (100411) |
| Mouse Ly-6G | APC | Rat | BioLegend (127613) |
| IgG2a isotype control | APC | Rat | BioLegend (400511) |
| Rabbit IgG | Alexa Fluor 647 | Goat | Abcam (ab150079) |
| Mouse IgG | Alexa Fluor 488 | Goat | ThermoFisher (A11001) |
| Iba1 | – | Rabbit | Wako (019–19741) |
| Neuron-specific $\beta$3 tubulin | – | Mouse | Abcam (ab78078) |
| P2X7 (extracellular) | – | Rabbit | Alomone Labs (APR-008) |
| Superoxide dismutase 1 | – | Sheep | Abcam (ab8866) |
| IgG isotype control | – | Rabbit | Abcam (ab171870) |
| IgG2a isotype control | – | Mouse | ICL (RS-90G2a) |
| Rabbit IgG | Peroxidase | Goat | Rockland (611-103-122) |
| Sheep IgG | Peroxidase | Donkey | Merck Millipore (AB324P) |

**Notes.**

[a] APC, allophycocyanin; Cy5.5, cyanine5.5; FITC, fluorescein isothiocyanate; Iba1, ionised calcium binding adaptor molecule 1; Ig, immunoglobulin; PE, phycoerythrin; PerCP, peridinin chlorophyll.

[b] Abcam, Cambridge, United Kingdom; AbD Serotec, Puchheim, Germany; Alomone Labs, Jerusalem, Israel; BioLegend, San Diego, CA; eBioscience, San Diego, CA; ICL, Immunology Consults Laboratory, Portland, OR; Merck Millipore, Billerica, MA; Rockland Immunochemicals, Gilbertsville, PA; ThermoFisher Scientific (Waltham, MA); Wako, Osaka, Japan.

## Animals

Animal studies were performed with the approval (AE11/29 and AE12/09) of the Animal Ethics Committee of the University of Wollongong (Wollongong, Australia). Mice hemizygous for the human SOD1$^{G93A}$ transgene and back-crossed (>5 generations) onto a C57BL/6J background (B6-Tg(SOD1-G93A)1GUr/j) were originally provided by Bradley Turner (University of Melbourne, Melbourne, Australia). Mice were bred and maintained at the Australian BioResources (ABR) Animal Facility (Moss Vale, Australia). At weaning, transgenic SOD1 mice were genotyped and relative SOD1 copy number assessed (Data S1). At 45–57 days of age (d), transgenic SOD1 mice with relatively similar SOD1 copy numbers were transported to and housed at the University of Wollongong in a temperature-controlled environment on a 12:12 h light-dark cycle, with light onset at 7:00 AM. Mice were matched for SOD1 copy number, age and sex, and divided into two treatment groups (Table 2). Mice were caged with paired littermates where possible, with 4–5 females or 3
**Table 2  Characteristics of mouse treatment groups at the beginning of treatment.**

| | Female | | Male | |
|---|---|---|---|---|
| | **Saline** | **BBG** | **Saline** | **BBG** |
| Number of mice ($n$) | 12 | 13 | 9 | 8 |
| Copy number ($\Delta Ct$) | $6.2 \pm 0.2$ | $6.2 \pm 0.1$ | $6.3 \pm 0.1$ | $6.3 \pm 0.1$ |
| Starting age (d) | $63.3 \pm 0.9$ | $63.5 \pm 0.8$ | $63.4 \pm 0.7$ | $63.6 \pm 0.7$ |
| Starting weight (g) | $17.2 \pm 0.8$ | $17.4 \pm 0.9$ | $22 \pm 1$ | $21.4 \pm 0.4$ |
| Starting ALS score | $0 \pm 0$ | $0 \pm 0$ | $0 \pm 0$ | $0 \pm 0$ |

**Notes.**

No significant differences between groups for any parameter (SOD1 transgene copy number, age, weight or score) at the beginning of treatment. Copy number is determined as the difference in the crossing points for the hSOD1 and mApoB curves ($\Delta Ct$). Results shown as means $\pm$ SD. BBG, Brilliant Blue G.

males per cage (unless fighting and required separation). Food and water was available *ad libitum*. When mice became symptomatic (clinical score > 2) and unable to access water or food, longer sippers were placed onto water bottles and food pellets placed directly onto the cage floor.

To investigate lumbar SOD1 and P2X7 proteins over time, female SOD1$^{G93A}$ mice or WT littermates lacking the SOD1 transgene at 30, 60, 90 and 120 d (3–5 per group) were genotyped by the Garvan Institute of Medical Research (Darlinghurst, Australia) mouse genotyping service. Mice were then processed at the University of Wollongong.

## Brilliant Blue G injections

BBG was diluted at 6 mg/mL in sterile 0.9% NaCl (saline). Both saline and the BBG preparation were filter-sterilised through 0.22 µm filters (Merck Millipore) and stored at −20 °C until required. Mice were injected intraperitoneally (i.p.) three times per week with BBG (45.5 mg/kg) from 62–64 d (i.e., prior to clinical disease onset) until end-stage, using Ultra-Fine 0.5 mL, 29G syringes (BD Biosciences, San Diego, CA, USA). Control littermates were injected i.p. with an equivalent volume of saline. Injections were performed on alternating sides of the abdomen to minimise irritation.

## Body weight measurements

Body weight was assessed three times a week prior to each injection using a Classic Plus PB602-S/M-FACT balance (Mettler Toledo, Columbus, OH, USA), with two initial measurements taken before the first injection, the average of which were defined as the pre-symptomatic disease maximum body weight (i.e., beginning at 58–60 d). Measurements were taken at the same time of day to minimise changes due to diurnal variations.

## ALS score

Mice were scored three times a week prior to each injection to assess neurological deficit, using the criteria outlined by the ALS Therapy Development Institute (Table 3). Scoring started at 58–60 d, with each score taken at the same time of day.

## Rotarod performance

Motor coordination was assessed weekly, beginning the first week of treatment (58–63 d), using a five-lane accelerating rotarod (RotaRod Advanced, TSE Systems, Hesse, Germany).

| Table 3 | Neurological scoring system. |
| --- | --- |
| **Score** | **Criteria** |
| 0 | Full extension of hind legs away from lateral midline when mouse is suspended by its tail; mouse can hold this for 2 s, suspended 2–3 times |
| 1 | Collapse or partial collapse of leg extension towards lateral midline (weakness) or trembling of hind legs during tail suspension |
| 2 | Toes curl under at least twice during walking of 12 inches, or any part of foot is dragging along cage bottom/table |
| 3 | Rigid paralysis or minimal joint movement, foot not being used for generating forward motion |
| 4 | Mouse cannot right itself within 30 s after being placed on either side |

Mice received two training sessions in the week prior to recording. These sessions consisted of one acclimatisation run with the rod rotating at a constant speed of 10 revolutions per minute (rpm), followed by two to three runs with the rod accelerating from 4–20 rpm over 180 s. Short rest intervals (minimum of 30 s) were given between runs. For testing, the rod was programmed to rotate at a constant speed of 4 rpm for 5 s to allow mice to acclimatise, followed by acceleration from 4–20 rpm over 180 s. The length of time mice could remain on the rod was recorded as the latency to fall, registered automatically by light-beam sensors under the rod. Mice were given at least three runs, with a minimum of 30 s between runs, to obtain 3 readings of at least 5 s. The maximum time each animal was able to remain on the rod each week was recorded and included in the data analysis. Mice that were unable to grip the rod due to advanced disease were given a minimum latency to fall of 5 s, the baseline time.

### Survival and end-stage

Transgenic SOD1$^{G93A}$ mice suffer from paralysis and muscle atrophy as disease progresses (*Gurney et al., 1994*). Thus, to minimise suffering, the disease end-stage was defined when mice first showed either (1) a 15% loss in body weight compared to their initial pre-symptomatic disease maximum body weight or (2) an inability to right themselves within 30 s after being placed on either side (i.e., a score of 4). Once the end-stage had been reached, mice were euthanised by asphyxiation using a slow-fill carbon dioxide inhalation technique.

Deceased animals with matching littermate controls were split into three post-study groups. For the first two groups, deceased mice were perfused with phosphate-buffered saline (PBS) followed by 4% PFA, or with PBS alone, and brains and spinal cords collected for immunohistochemical and biochemical analyses, respectively. For the third group, serum, spleens and livers were collected from non-perfused animals for further analysis.

### Immunohistochemistry
#### Spinal cord processing

Immediately following euthanasia, mice were transcardially perfused with PBS followed by 4% PFA once perfusate ran clear. Spinal cords were then dissected, post-fixed with 4% PFA at RT for 2 h, and then washed twice with PBS. Following this, spinal cords were cryoprotected in 30% sucrose in PBS at 4 °C for a minimum of 2 d. Tissue was embedded in OCT medium, frozen by suspending above a layer of liquid nitrogen, and stored at −80 °C (maximum of 10 d). Spinal cord transverse sections of 20 μm thickness were cut from the

lumbar regions of the spinal cords on a CM1950 cryostat (Leica, Mannheim, Germany), using spinal cord enlargement to identify the lumbar region. Sections were mounted onto StarFrost® advanced adhesive slides (Knittel Glaser, Braunschweig, Germany) and stored at −80 °C (maximum of four weeks).

### Iba1 and β3-tubulin staining

Microglia and motor neuron numbers were assessed by staining mounted spinal cords for Iba1 and $\beta$3-tubulin, respectively. Firstly, a pap pen (Daido Sangyo, Tokyo, Japan) was used to separate tissue sections on the same slide. All incubations were carried out in a humidified chamber. Sections were fixed with 4% PFA in PBS at room temperature (RT) for 15 min, and then washed three times with PBS over 30 min. Sections were then blocked with 20% NHS in PBS at RT for 20 min, and incubated at 4 °C overnight with rabbit anti-Iba1 polyclonal antibody (pAb) (0.1 µg/100 µL) and mouse $\beta$3-tubulin monoclonal antibody (mAb) (0.5 µg/100 µL), or rabbit IgG isotype control pAb and mouse $IgG_{2a}$ isotype control at corresponding concentrations, all in PBS containing 1% BSA, 0.2% NHS and 0.05% $NaN_3$. The following day, sections were washed as above, and incubated at RT for 1 h with Alexa Fluor 647-conjugated goat anti-rabbit IgG Ab (0.3 µg/100 µL) and Alexa Fluor 488-conjugated goat anti-mouse IgG Ab (0.3 µg/100 µL) in PBS containing 0.2% NHS. Cells were washed as above and then coverslips mounted onto tissue sections with 50% (v/v) glycerol gelatin in PBS. Coverslips were sealed with nail varnish. Sections were visualised using a TCS SP5 II confocal imaging system and images of anterior horns captured using Leica Application Suite Advanced Fluorescence Lite software (version 2.6.3) (Leica) (excitation 633, emission collected at 655–695 nm for Iba1; excitation 488 nm, emission collected at 510–550 nm for $\beta$3-tubulin). Images were taken of 6–12 anterior horns per animal, for a total of 12 animals (6 per treatment group). Spinal cord anterior horns were defined as the regions of gray matter on the ventral side of a horizontal line crossing through the central canal.

### Motor neuron counts

The numbers of motor neurons in anterior horns were manually counted using images of $\beta$3-tubulin-stained spinal cords visualised in Leica Application Suite Advanced Fluorescence Lite software. Motor neurons were defined as cells with cell body diameters of at least 15 µm showing positive immunostaining for $\beta$3 tubulin. Counts from the right and left anterior horns for each section were averaged, and then the average count from 3–6 sections used as a single measurement per animal for statistical analyses.

### Microglia density

Microglia density was assessed in anterior horns using images of Iba1-stained spinal cords and ImageJ software (version 1.48) (National Institutes of Health, Bethesda, MD). Firstly, thresholds were adjusted into 16-bit black and white images to account for any differences in the intensity of staining or in the background fluorescence between sections. Particles between 10–1,000 pixel units were then counted, and the percent area showing positive immunostaining for Iba1 of the total anterior horn calculated. Percent areas from the right and left anterior horns for each section were averaged, and then the average count from 3–6 sections used as a single measurement per animal for statistical analyses.

## Immunoblotting

### Spinal cord protein extraction

Immediately following euthanasia, mice were transcardially perfused with PBS. Spinal cords were then dissected, and the lumbar portion excised using spinal cord enlargement to identify the region. Lumbar spinal cord segments were snap-frozen in liquid nitrogen and stored at −80 °C until required. Detergent-soluble proteins were extracted from lumbar spinal cords by homogenising tissue in ice-cold neuronal protein extraction reagent containing 100 × Halt$^{TM}$ protease inhibitor cocktail using a micropestle. A ratio of 10 µL extraction reagent per 1 mg of tissue was utilised. Homogenates were incubated on ice for 10 min and cleared (20,000 × g at 4 °C for 10 min). To extract detergent-insoluble proteins, the resulting pellet was resuspended in 0.5 M Tris HCl (pH 6.8) containing 2% w/v SDS at the same volume as utilised for soluble protein extraction (to maintain a consistent concentration with the original solution), and incubated at RT for 10 min. Unsolubilised material was cleared (20,000 × g for 15 min). Soluble and insoluble protein samples were stored at −20 °C.

### P2X7 and SOD1 protein detection

Soluble protein (50 µg) or an equivalent volume of insoluble protein were separated under reducing conditions (5% $\beta$-mercaptoethanol) using Any kD Mini-PROTEAN TGX Stain-Free Gels (Bio-Rad, Hercules, CA, USA). Before immunoblotting, equal protein loading was confirmed by visualising stain-free gels using a Bio-Rad Criterion Stain Free Imager and Image Lab software. Proteins were then transferred to nitrocellulose membranes (Bio-Rad) using a Bio-Rad Trans-Blot Turbo Transfer System. To allow for separate P2X7 and SOD1 immunoblotting, membranes were cut horizontally between 25 and 37 kDa, using the pre-stained marker as a guide. Both halves were then blocked at RT for 1 h with Tris-buffered saline (250 mM NaCl and 50 mM Tris, pH 7.5) containing 0.2% Tween-20 and 5% milk powder, and then incubated at 4 °C overnight with either an anti-P2X7 pAb (1:500) or anti-SOD1 pAb (1:1000) in Tris-buffered saline containing 0.2% Tween-20 and 5% milk powder. Membranes were washed three times over 30 min with Tris-buffered saline containing 0.2% Tween-20. Membranes were then incubated at RT for 1 h with peroxidise-conjugated anti-rabbit (1:1000) or anti-sheep (1:500) IgG Ab (for P2X7 or SOD1, respectively) in Tris-buffered saline containing 0.2% Tween-20 and 5% milk powder. Membranes were washed as above, and proteins visualised using chemiluminescent substrate and Amersham Hyperfilm ECL (GE Healthcare, Little Chalfont, Buckinghamshire, UK). Films were processed using GBX Developer and Replenisher, and GBX Fixer and Replenisher as per the manufacturer's instructions (Kodak Australasia, Collingwood, Australia). Images of films were collected using a GS-800 Calibrated Densitometer (Bio-Rad). Relative P2X7 and SOD1 was quantified from these images using ImageJ software, normalising to a single soluble protein sample included on all gels.

## Serum monocyte chemoattractant protein-1 measurements

Blood was collected immediately following euthanasia via cardiac puncture using a 27G PrecisionGlide needle (BD Biosciences). Blood was incubated at RT for 45–90 min and then centrifuged (1,700 × g for 5 min). Serum was stored at −80 °C. The amount of MCP-1

present in serum was measured using a mouse CCL2 (MCP-1) ELISA Ready-SET-Go! kit (eBioscience, San Diego, CA, USA), as per the manufacturer's instructions.

## Splenic leukocyte phenotyping

Spleens from euthanised mice were homogenised in PBS and filtered through Falcon 70 μm nylon filters (BD Biosciences). Cells were washed with PBS ($300 \times$ g for 5 min), resuspended in lysing buffer (155 mM $NH_4Cl$, 10 mM $KHCO_3$ and 0.1 mM $Na_2EDTA$, pH 7.4), and incubated at RT for 5 min. Cells were then washed as above and resuspended in cold PBS containing 2% FBS. Cells ($1 \times 10^6$/tube) were incubated on ice for 20 min with combinations of fluorochrome-conjugated anti-mouse mAbs or corresponding isotype control mAbs. Cells were washed as above, and events collected using a LSRFortessa X-20 Cell Analyzer (BD Biosciences) (excitation 488 nm, emission collected with 525/50 and 695/40 band-pass filters for FITC- and PerCP/Cy5.5-conjugated mAbs, respectively; excitation 561, emission collected with 585/15 band-pass filter for PE-conjugated mAbs; excitation 640, emission collected with 670/30 band-pass filter for APC-conjugated mAbs). Percentages of leukocyte subsets were determined using FlowJo software (version 8.7) (Tree Star, Ashland, OR, USA).

## P2X7 expression by qPCR

Sections of livers and spleens, approximately 5 $mm^3$ in size, were stored in RNAlater at 4 °C. After 1–2 d, tissues were transferred to −20 °C. Tissues were removed from RNAlater and total RNA isolated using TRIzol reagent (ThermoFisher Scientific), as per the manufacturer's instructions. cDNA was synthesised from isolated RNA using qScript cDNA Synthesis Kit (Quanta Biosciences, Gaithersburg, MD, USA), as per the manufacturer's instructions. qPCR amplification was performed in a 10 μL reaction volume, containing 5 μL $2 \times$ TaqMan Universal Master Mix II (ThermoFisher Scientific), 2 μL liver or spleen cDNA, 0.5 μL of $20 \times$ TaqMan Gene Expression Assay-Specific primers/probes (ThermoFisher Scientific), and PCR-grade water. Standard primers for murine *P2RX7* (FAM-labelled; Mm00440578_m1) and murine glyceraldehyde 3-phosphate dehydrogenase (*GAPDH*) (VIC-labelled; Mm99999915_g1) were used. Amplification was performed using an Eco Real-Time PCR System (Illumina, San Diego, CA, USA). Reactions were held at 50 °C for 2 min and 95 °C for 10 min, followed by 40 two-segment cycles of 95 °C for 15 s and 60 °C for 1 min. A single fluorescence acquisition was taken each cycle, using FAM (505–545 nm) and VIC (562–596 nm) filter combinations. No-template controls were included in each run and all samples were run in triplicate. cDNA obtained from the spleen of a BALB/c mouse (*Watson et al., 2014*) was also included in each run to compare relative P2X7 expression (given a value of 1). Relative gene expression was normalised to the murine GAPDH housekeeping gene and determined using EcoStudy software (version 4.1.2.0) (Illumina).

## Data presentation and statistical analyses

Data is presented as mean $\pm$ SD, unless otherwise indicated. Differences between multiple treatments were compared by ANOVA paired with Tukey's HSD post-tests. For single comparisons, unpaired Student's *t*-tests were performed. Differences in body weight between treatment groups were compared using an unpaired Student's *t*-tests in two ways: first, for the mean percentage body weight for each mouse over the entire period of the

study; second, for the percentage body weight for each mouse at each recorded time-point. For ALS score, rotarod performance and survival data, differences in the values of BBG- and saline-treated mice were assessed using Kaplan–Meier analysis paired with log-rank tests, considering the time to reach a consistent score of 2, drop to a consistent run of less than 60 s or end-stage, respectively, as an event. To prevent decomposition of means due to animals reaching end-stage, for weight, score and rotarod analyses, the last values prior to euthanasia were carried forward until the last mouse in each group had reached end-stage. These values were used to compute means at the end of the study. The software package Prism 5 for Windows (version 5.01) (GraphPad Software, San Diego, CA, USA) was used for all statistical analyses, with differences considered significant for $P < 0.05$.

## RESULTS

### Brilliant Blue G treatment reduces body weight loss in SOD1$^{G93A}$ mice, but has no effect on clinical score, motor coordination or survival

The P2X7 antagonist, BBG, has been used in over 20 pre-clinical studies of neurological-related disease in rodents (*Bartlett, Stokes & Sluyter, 2014*). To investigate the therapeutic efficacy of BBG treatment on ALS progression, SOD1$^{G93A}$ mice were treated with saline or BBG from 62–64 d (i.e., prior to clinical disease onset) until end-stage, and body weight, ALS score and motor coordination recorded for the duration of the study.

Body weight of saline- and BBG-treated SOD1$^{G93A}$ mice was maintained or increased until approximately 105 d, after which continuous body weight loss was observed (Fig. 1A). Overall, BBG-treated mice had greater initial weight gain and a delayed decline in weight compared to saline-treated mice, in addition, BBG treated mice had a higher mean percentage body weight ($P = 0.014$, $n = 21$), when calculated for each mouse over the entire period of the study (Fig. 1A, insert). Furthermore, at 63, 66, 80, 82, 84, 87, 89, 91, 94, 96, 98, 101, 103, 105, 108, 117, 119, 122 and 131 d, BBG-treated mice had significantly greater percentage body weights compared to saline-treated mice ($P < 0.05$, $n = 21$) (Fig. 1A).

To assess neurological deficit, mice were scored from 58-60 d using the criteria outlined by the ALS Therapy Development Institute (Table 3). At 58–60 d, all saline- and BBG-treated mice had clinical scores of 0 (Fig. 1B). These scores increased throughout the study, plateauing at an approximate score of 2.5 at 145 and 150 d for saline- and BBG-treated mice, respectively. Overall, considering the time to maintain a consistent score of 2 as an event, there was no significant difference in score between saline- and BBG-treated mice ($P = 0.349$, $n = 21$) (Fig. 1B).

Motor coordination was assessed weekly using an accelerating rotarod, beginning at 58–63 d. Motor coordination initially rose for both saline- and BBG-treated mice, consistent with mice continuing to learn how to use the rod, and was then maintained with some variation until 95 d (Fig. 1C). After 95 d, motor coordination rapidly declined for both treatment groups, with comparable rates of decline ($P = 0.311$, $n = 21$ considering the time to reach a latency to fall consistently less than 60 s) (Fig. 1C).

End-stage was considered reached once mice displayed either a 15% loss in body weight compared to their initial pre-symptomatic disease maximum body weight, or an inability

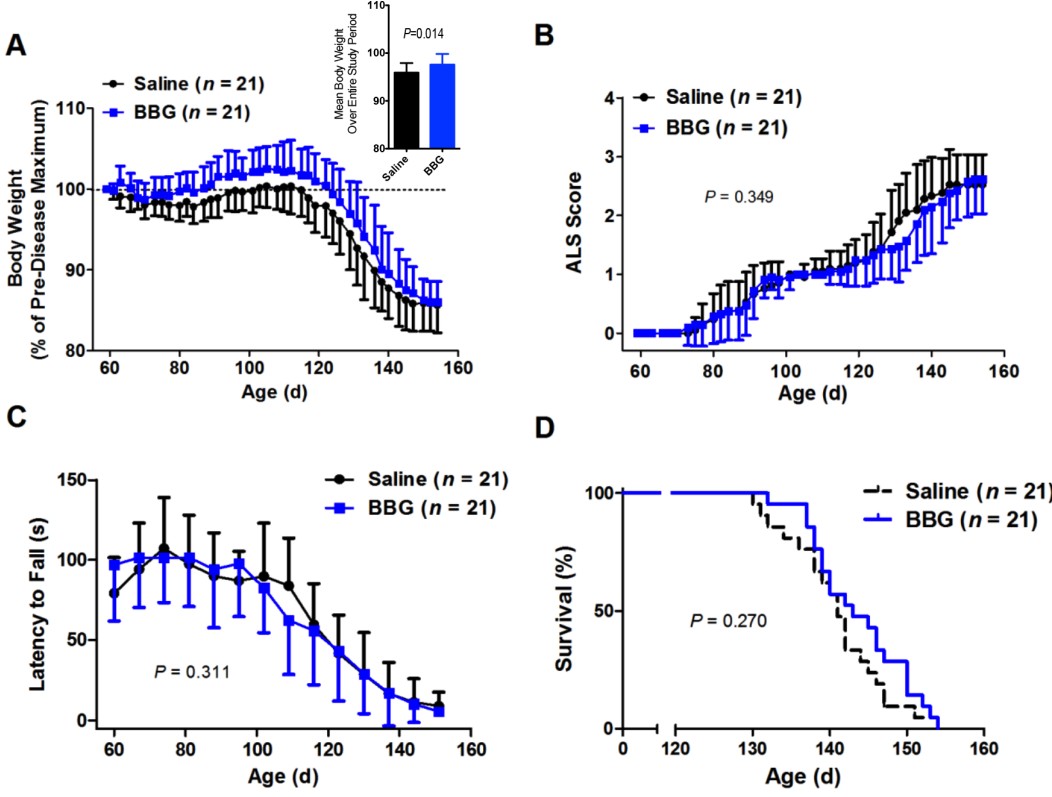

**Figure 1 Brilliant Blue G (BBG) treatment reduces body weight loss in SOD1$^{G93A}$ mice.** SOD1$^{G93A}$ mice were injected i.p. with 45.5 mg/kg BBG or an equivalent volume of saline 3 times per week from 62–64 d until end-stage. During this period, (A) body weight, inset shows the mean body weight over the entire experiment, (B) neurological deficit and (C) motor coordination were assessed (A–B) three times or (C) once per week using (B) the ALS score criteria outlined by the ALS Therapy Development Institute or (C) an accelerating rotarod programmed to rotate at a constant acclimatising speed of 4 rpm for 5 seconds (s), followed by acceleration from 4–20 rpm over 180 s. (D) Mice were euthanased once either a 15% loss in body weight compared to initial pre-symptomatic disease maximum body weight was reached or an inability to right within 30 s after being placed on either side was demonstrated. Results are shown as the (A–C) mean (A) percentages of the pre-disease (asymptomatic) maximum body weight, (B) ALS score or (C) latency to fall (A–C) ±SD or (D) percent survival for BBG- and saline-treated mice. Comparisons between saline- and BBG-treated mice were made using (A) an unpaired Student's *t*-test on average weight of mice over the entire period or (B–D) log-rank tests.

to right themselves within 30 s after being placed on either side (i.e., a score of 4). BBG treatment, beginning at 62–64 d, did not affect the survival rate of SOD1$^{G93A}$ mice (saline 141 ± 6 d vs. BBG 144 ± 6 d, $P = 0.270$, $n = 21$), with similar rates of decline in percent survival observed for both saline- and BBG-treated mice (Fig. 1D). However, there appeared to be a tendency for this rapid decline to be delayed in BBG-treated mice (Fig. 1D).

## Brilliant Blue G treatment reduces body weight loss and prolongs survival in female but not male SOD1$^{G93A}$ mice, but has no effect on clinical score or motor coordination in either sex

The effects of P2X7 antagonism in a mouse model of ALS have previously been reported to be gender-dependent (*Cervetto et al., 2013*). Thus, subgroup analyses by gender were

performed to elucidate whether any differences in body weight, clinical score, motor coordination or survival could be observed following BBG treatment in either males or females.

Body weight loss occurred in a comparable manner for both saline- and BBG-treated male mice ($P = 0.649$, $n = 8$–$9$) (Fig. 2A and insert). Furthermore, there were no significant differences in male body weight as a percentage of the pre-symptomatic disease maximums on any weight measurement day. In contrast, BBG-treated female mice had significantly greater initial weight gain and a delayed decline in weight compared to saline-treated female mice over the entire period of the study ($P = 0.006$, $n = 12$–$13$) (Fig. 2A and insert). At 63, 66, 73, 80, 82, 84, 87, 89, 91, 94, 96, 98, 101, 103, 105, 108, 117, 119, 122, 126 and 131 d, BBG-treated female mice had significantly greater percentage body weights compared to saline-treated female mice ($P < 0.05$, $n = 12$–$13$) (Fig. 2A).

Score trajectories similar to those described above were observed for both male and female saline- and BBG-treated mice (Fig. 2B). Overall, there were no significant differences in score between saline- and BBG-treated male or saline- and BBG-treated female mice when considering the time to maintain a consistent score of 2 as an event (male $P = 0.647$, $n = 8$–$9$; female $P = 0.084$, $n = 12$–$13$) (Fig. 2B).

Similar patterns of motor deficit to that described above were also observed for both male and female saline- and BBG-treated mice (Fig. 2C), again with comparable rates of decline between those treated with saline and those with BBG (male $P = 0.390$, $n = 8$–$9$; female $P = 0.586$, $n = 12$–$13$; considering the time to reach a latency to fall consistently less than 60 s).

When considering males alone, survival was not affected by BBG treatment (saline $143 \pm 7$ d vs. BBG $142 \pm 6$ d, $P = 0.582$, $n = 8$–$9$) (Fig. 2D). Conversely, for females, BBG treatment extended survival (saline $139 \pm 6$ d vs. BBG $145 \pm 6$ d, $P = 0.035$, $n = 12$–$13$) (Fig. 2D). Whilst similar rates of decline were observed for both female saline- and BBG-treated mice, this decline was delayed following BBG treatment in females.

## Brilliant Blue G treatment does not reduce motor neuron loss or microgliosis in SOD1$^{G93A}$ mice

To determine the effect of BBG-treatment on motor neuron loss and microgliosis, neurons and microglia in the anterior horn of lumber spinal cords from end-stage saline- and BBG-treated mice were labelled and imaged by confocal microscopy. Motor neurons were counted and the density of microglia assessed. The lumbar spinal cord was chosen as this is the CNS tissue in which lower motor neurons reside and that is predominately damaged during ALS (*Apolloni et al., 2013a*).

Representative images of stained anterior horn sections from saline and BBG-treated SOD1$^{G93A}$ mice are shown (Fig. 3). A stained section derived from a WT littermate of corresponding age is included for comparison. The average number of motor neurons, identified as cells with cell body diameters of at least 15 μm, was similar between saline- and BBG-treated SOD1$^{G93A}$ mice (saline $19.3 \pm 2.0$ vs. $19.5 \pm 1.4$ motor neurons, $P = 0.868$, $n = 6$) (Fig. 3A). Similarly, there was no difference in the amount of microgliosis between saline- and BBG-treated mice (saline $2.6 \pm 0.3\%$ vs. $2.5 \pm 0.8\%$, $P = 0.821$, $n = 6$) (Fig. 3B).

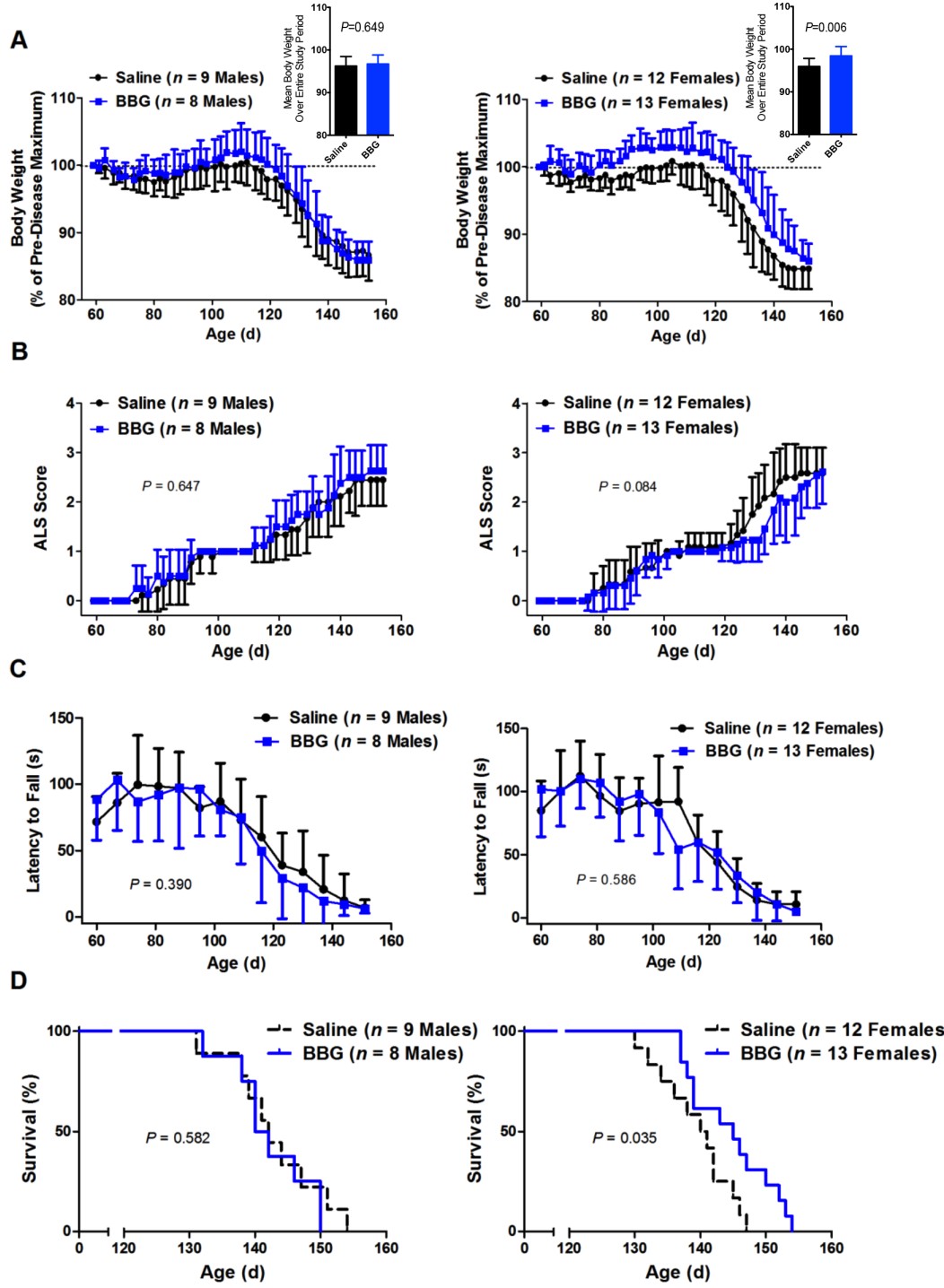

**Figure 2** **Brilliant Blue G (BBG) treatment reduces body weight loss and prolongs survival in female, but not male, SOD1$^{G93A}$ mice.** SOD1$^{G93A}$ mice from Fig. 1 were stratified according to sex. Results are shown as the (A–C) mean (A) percentages of the pre-disease (asymptomatic) maximum body weight, inset shows the mean body weight over the entire experiment, (B) ALS score or (C) latency to fall (A–C) ±SD or (D) percent survival for male or female BBG- and saline-treated mice. Comparisons between saline- and BBG-treated mice were made using (A) unpaired Student's $t$-tests over the entire period or (B–D) log-rank tests.

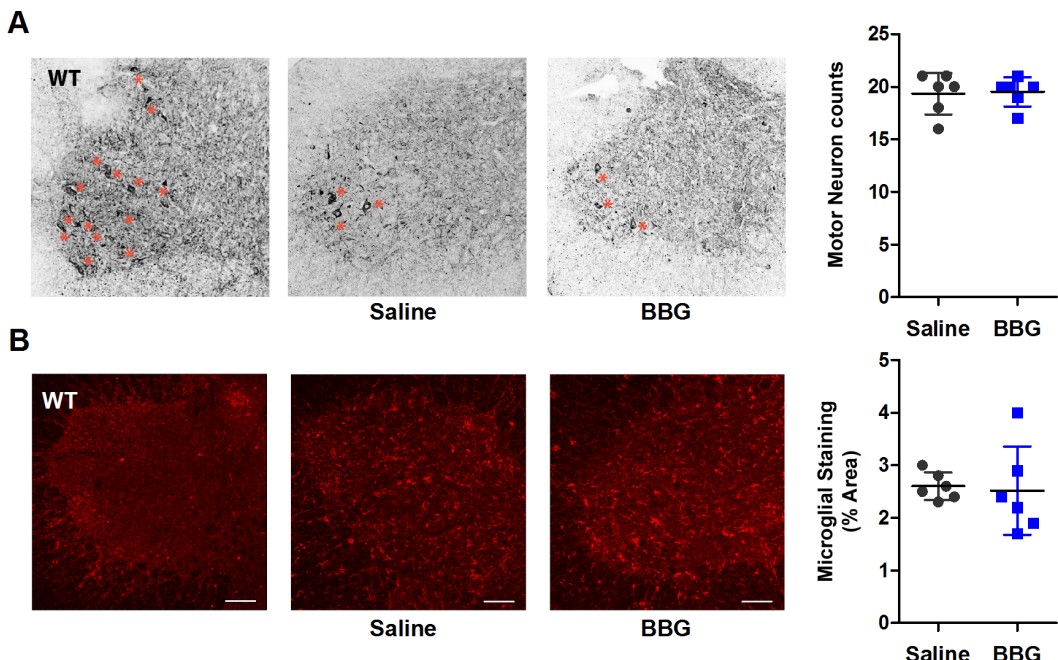

**Figure 3** **Brilliant Blue G (BBG) treatment does not reduce motor neuron loss or microgliosis in the anterior horn of SOD1$^{G93A}$ mice.** Lumbar spinal cord transverse cross sections (20 μm) from BBG- or saline-treated SOD1$^{G93A}$ mice at end-stage were fixed with paraformaldehyde and labelled with an (A) anti-β3-tubulin mAb and Alexa Fluor 488-conjugated anti-IgG Ab or (B) anti-Iba1 pAb and Alexa Fluor 647-conjugated anti-IgG Ab, and (A, B) analysed by confocal microscopy. Representative confocal images of anterior horn are shown. Bars represent 100 μm. A stained section derived from a single wild type (WT) littermate of corresponding age is shown for comparison. Results shown as mean (A) motor neuron (neuron ≥ 15 μm in diameter) count per anterior horn or (B) percent area positive for Iba-1 (A, B) ±SD, $n = 6$ SOD1$^{G93A}$ mice (male and female). Symbols represent individual animals.

## Brilliant Blue G treatment does not affect the amount of detergent-soluble or -insoluble lumbar SOD1 or P2X7 protein in SOD1$^{G93A}$ mice

To determine whether prolonged BBG treatment reduced the amount of P2X7 and SOD1 in the lumbar spinal cords of end-stage SOD1$^{G93A}$ mice, detergent-soluble and -insoluble lumbar spinal cord protein fractions were examined by immunoblotting and densitometry. All samples were normalised to one soluble sample (indicated in red). Monomeric SOD1 (16 kDa) was present in both soluble and insoluble fractions, with similar amounts detected between saline- and BBG-treated mice (soluble $P = 0.842$, $n = 7$; insoluble $P = 0.149$, $n = 7$) (Figs. 4A and 4C).

Immunoblotting with a P2X7 pAb raised against an extracellular epitope revealed multiple proteins, with approximate sizes of 155, 60 and 50 kDa, as well as a doublet around 73–89 kDa (Figs. 4B and 4D). This doublet most likely corresponds to full-length P2X7 monomers (P2X7A) (*Masin et al., 2012*). There was no difference in total P2X7 amounts between saline- and BBG-treated mice in either soluble or insoluble fractions (soluble $P = 0.821$, $n = 7$; insoluble $P = 0.776$, $n = 7$) (Figs. 4B and 4D). There was also no difference in amounts of P2X7A between saline- and BBG-treated mice in either soluble or insoluble fractions (soluble $P = 0.571$, $n = 7$; insoluble $P = 0.689$, $n = 7$) (Figs. 4B and 4D).

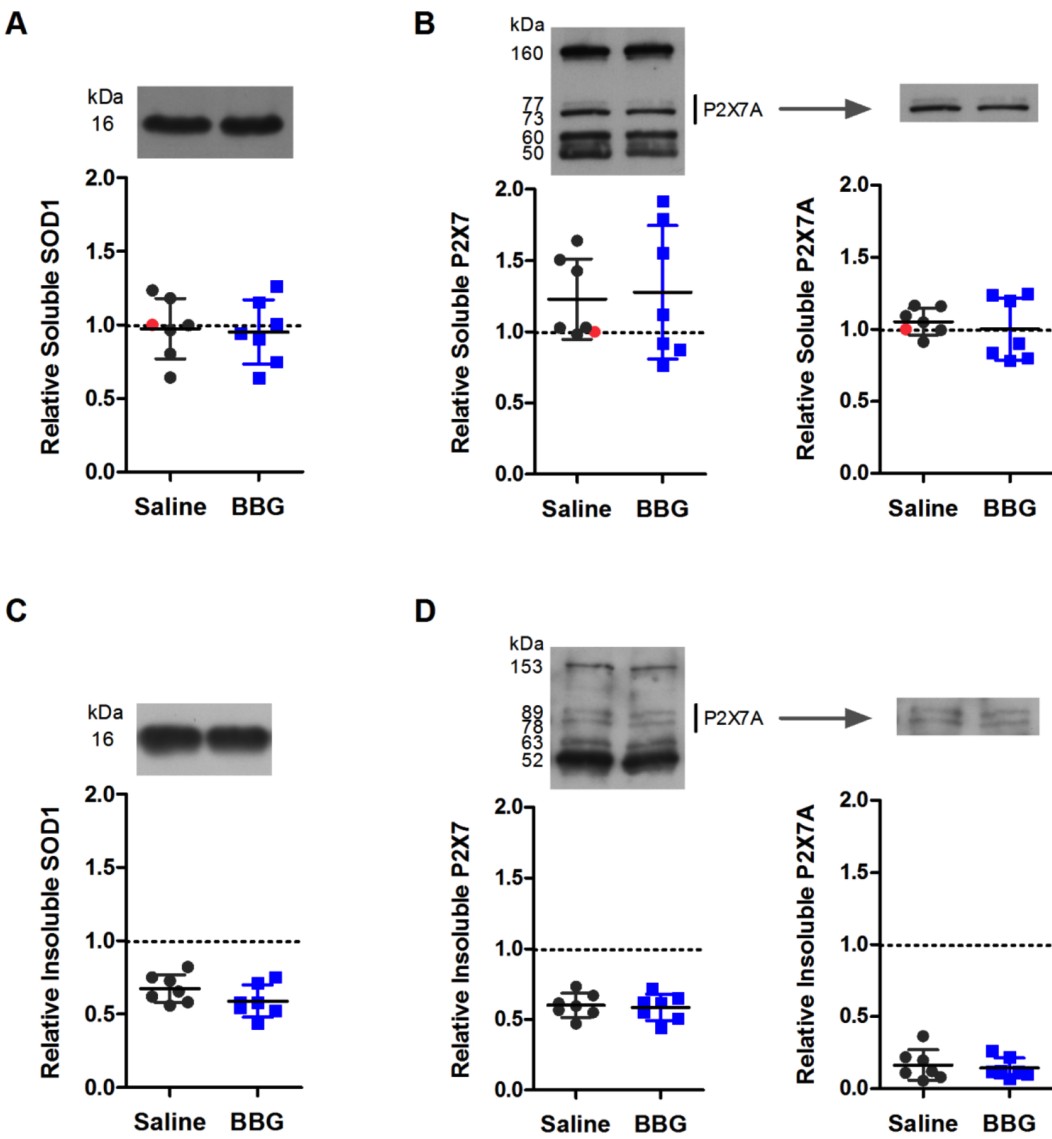

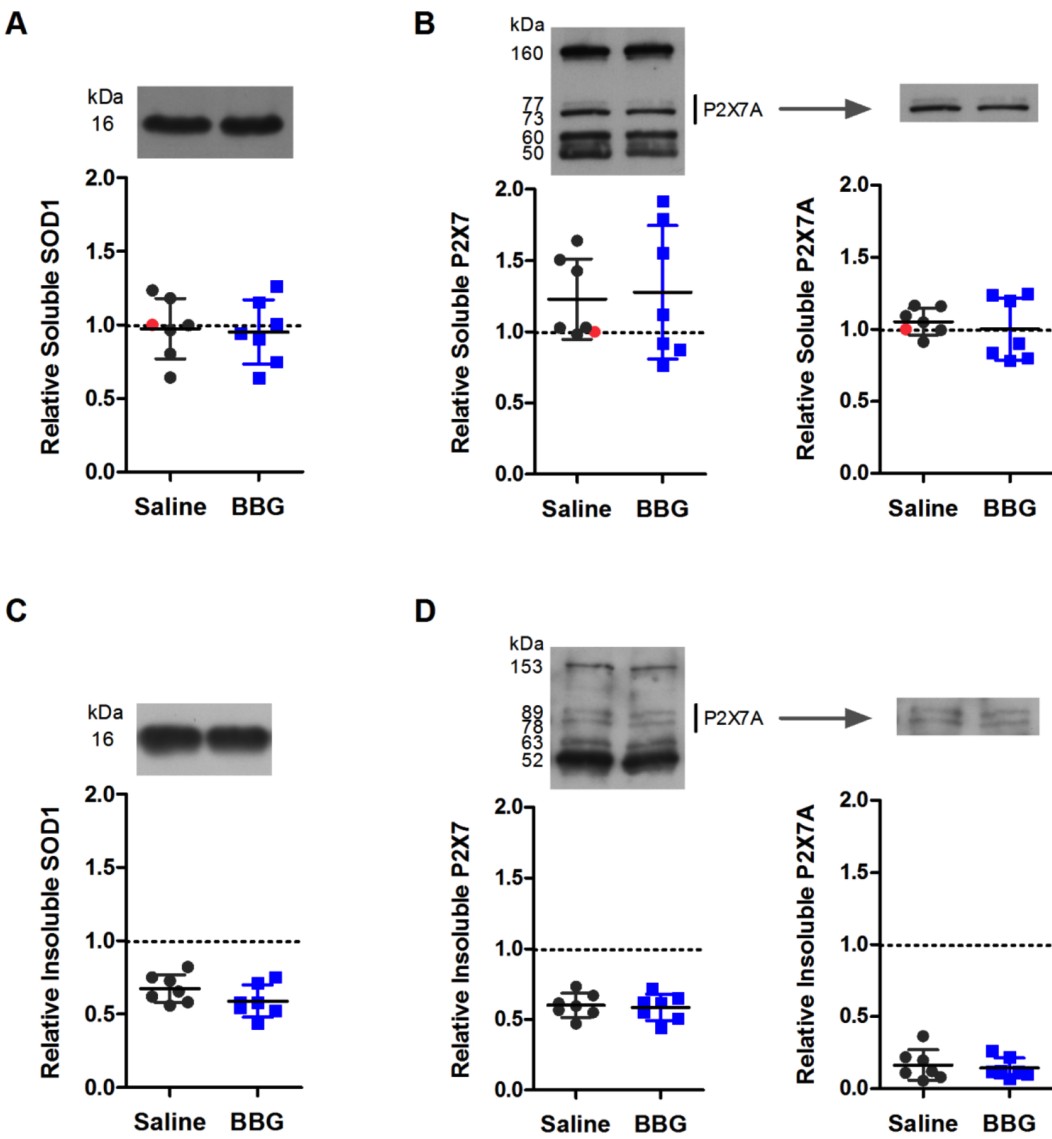

**Figure 4 Brilliant Blue G (BBG) treatment does not affect the amount of detergent-soluble or -insoluble lumbar SOD1 or P2X7 protein in SOD1[G93A] mice.** (A, B) Soluble or (C, D) insoluble protein fractions from homogenised BBG- or saline-treated SOD1[G93A] mouse lumbar spinal cords at end-stage were separated by SDS-PAGE under reducing conditions, transferred to nitrocellulose membranes, and incubated with (A, C) anti-SOD1 or (B, D) anti-P2X7 pAbs. Representative blots are indicated. Results shown as mean (A, B) soluble or (C, D) insoluble SOD1 or P2X7 protein $\pm$ SD, $n = 7$ mice (male and female). Amounts shown are relative to corresponding soluble proteins of one saline mouse (indicated in red). Symbols represent individual animals.

## Brilliant Blue G treatment partially increases serum MCP-1 concentrations in SOD1[G93A] mice

MCP-1, an inflammatory chemokine involved in the recruitment of monocyte-derived cells such as microglia, is up-regulated in ALS mice (*Kawaguchi-Niida et al., 2013*) and patients (*Henkel et al., 2004*; *Nagata et al., 2007*; *Wilms et al., 2003*). Thus, the concentrations of MCP-1 in the serum of end-stage saline- and BBG-treated mice was investigated to

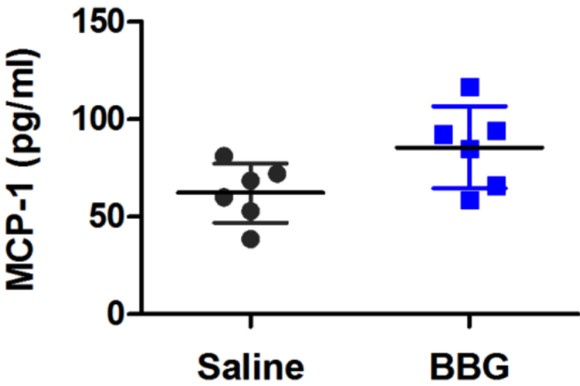

**Figure 5** **Brilliant Blue G (BBG) treatment partially increases serum monocyte chemoattractant protein-1 (MCP-1) concentrations in SOD1$^{G93A}$ mice.** Serum from BBG- or saline-treated SOD1$^{G93A}$ mice at end-stage was collected and MCP-1 concentrations determined using an ELISA. Results shown as means $\pm$ SD, $n = 6$ (male and female). Symbols represent individual animals.

determine whether BBG treatment was able to reduce serum MCP-1. Low concentrations of MCP-1 were detected in both saline- and BBG-treated mice (Fig. 5). In contrast to expectations, serum MCP-1 concentrations were 1.4-fold higher in BBG-treated mice compared to saline-treated mice; a difference approaching statistical significance ($P = 0.052$, $n = 6$).

## Brilliant Blue G treatment does not affect the distribution of splenic leukocytes from SOD1$^{G93A}$ mice

The percentage of splenic leukocyte subsets of total CD45$^+$ leukocytes from end-stage saline- and BBG-treated SOD1$^{G93A}$ mice was investigated by flow cytometry. There were no differences in the percentages of B cells (CD19$^+$CD3$^-$; $P = 0.362$, $n = 6$), total T cells (CD3$^+$CD19$^-$; $P = 0.615$, $n = 6$), CD8$^+$ T cells (CD8$^+$CD3$^+$CD19$^-$; $P = 0.643$, $n = 6$), CD4$^+$ T cells (CD4$^+$CD3$^+$; $P = 0.549$, $n = 6$), regulatory T cells (CD25$^+$CD4$^+$CD3$^+$; $P = 0.239$, $n = 6$), dendritic cells (DCs; CD11b$^\pm$CD11c$^+$; $P = 0.981$, $n = 6$), macrophages (MOs; CD11b$^+$CD11c$^-$ Ly6G$^-$; $P = 0.130$ $n = 6$) or polymorphonuclear neutrophils (PMNs; CD11b$^+$CD11c$^-$ Ly6G$^+$; $P = 0.084$, $n = 6$) between saline- and BBG-treated mice (Fig. 6).

## Brilliant Blue G treatment does not affect P2X7 expression in the spleens or livers of SOD1$^{G93A}$ mice

To determine if the prolonged BBG treatment altered P2X7 expression in non-CNS tissues, the murine P2X7 expression, normalised to murine GAPDH, in spleens and livers from end-stage saline- and BBG-treated mice was determined by qPCR. Relative P2X7 expression in either the spleens ($P = 0.644$, $n = 6$) or livers ($P = 0.942$, $n = 6$) was similar between saline- and BBG-treated mice (Fig. 7).

## The amount of detergent-soluble or -insoluble lumbar SOD1 or P2X7 protein does not change over time in female SOD1$^{G93A}$ mice

To determine whether the amount of SOD1 or P2X7 protein in the lumbar spinal cords of SOD1$^{G93A}$ mice change with age, and whether differences exist in P2X7 protein levels

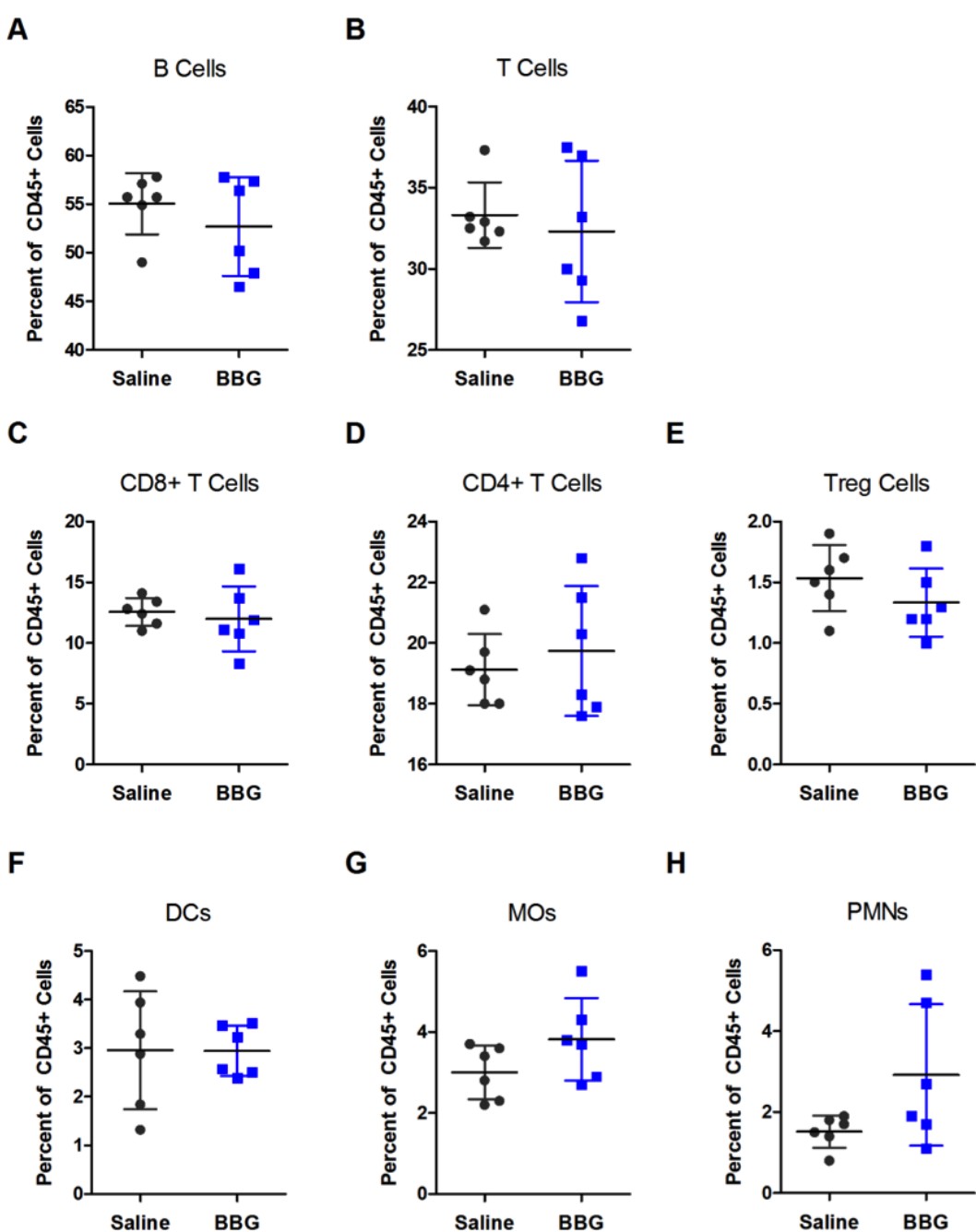

**Figure 6** **Brilliant Blue G (BBG) treatment does not affect the distribution of splenic leukocytes from SOD1^{G93A} mice.** Splenic leukocytes from BBG- or saline-treated SOD1$^{G93A}$ mice at end-stage were labelled with fluorochrome-conjugated mAbs. The percentage of (A) B cells (CD19$^+$CD3$^-$), (B) T cells (CD3$^+$CD19$^-$), (C) CD8$^+$ T cells (CD8$^+$CD3$^+$CD19$^-$), (D) CD4$^+$ T cells (CD4$^+$CD3$^+$), (E) regulatory T (Treg) cells (CD25$^+$CD4$^+$CD3$^+$), (F) dendritic cells (DCs; CD11b$^{\pm}$CD11c$^+$), (G) macrophages (MOs; CD11b$^+$CD11c$^-$Ly6G$^-$) and (H) polymorphonuclear neutrophils (PMNs; CD11b$^+$CD11c$^-$Ly6G$^+$) of the total CD45$^+$ leukocyte population were determined by flow cytometry. Results shown as means $\pm$ SD, $n = 6$ (male and female). Symbols represent individual animals.

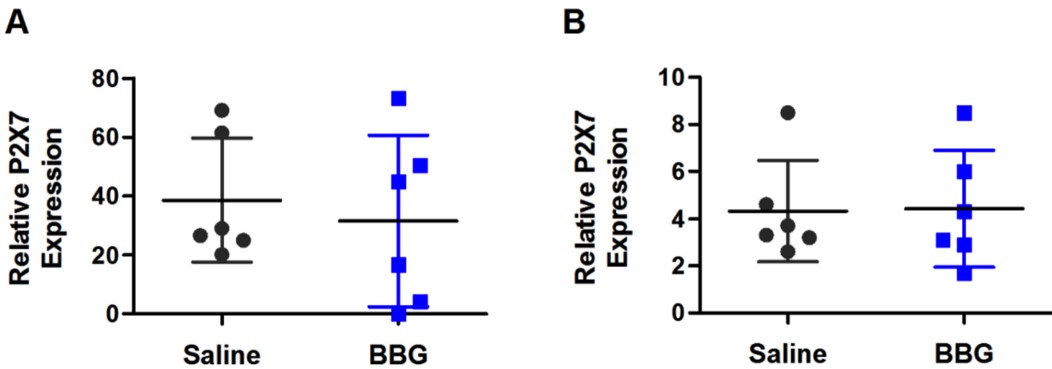

**Figure 7  Brilliant Blue G (BBG) treatment does not affect P2X7 expression in the spleens or livers of SOD1$^{G93A}$ mice.** RNA was isolated from (A) spleens and (B) livers from BBG- or saline-treated SOD1$^{G93A}$ mice at end-stage. cDNA was synthesised from isolated RNA and amplified by qPCR using probes for P2X7 and GAPDH to determine P2X7 expression (normalised to GAPDH). P2X7 expression is relative to one control sample (obtained from BALB/c spleen). Results shown as means ± SD, $n = 6$ (male and female). Symbols represent individual animals.

between SOD1$^{G93A}$ mice and their WT littermates lacking the SOD1 transgene, detergent-soluble and -insoluble lumbar spinal cord fractions were prepared and proteins examined by immunoblotting. Given that BBG treatment delayed weight loss and prolonged survival in females, but not males (Fig. 2), female mice were used for these analyses. Human SOD1 (16 kDa) was present in both soluble and insoluble fractions derived from SOD1$^{G93A}$ mice, but was not detected in fractions derived from WT non-transgenic littermates (Figs. 8A and 8B). The amount of detergent-soluble and -insoluble SOD1 did not significantly change over time in lumbar spinal cords of SOD1$^{G93A}$ mice (soluble $P = 0.160$, $n = 4–5$; insoluble $P = 0.794$, $n = 4–5$).

Similar to previous observations (Fig. 4), immunoblotting with a P2X7 Ab raised against an extracellular epitope revealed multiple proteins, with approximate sizes of 160, 60 and 50 kDa, as well as a doublet around 80–90 kDa (Figs. 8C and 8D). Total P2X7 amounts were similar between fractions derived from WT and SOD1$^{G93A}$ mice at each time point. The amount of detergent-soluble or -insoluble total P2X7 did not significantly change over time in lumbar spinal cords of WT or SOD1$^{G93A}$ mice (SOD1 soluble $P = 0.144$, $n = 4–5$; WT soluble $P = 0.292$, $n = 3–4$; SOD1 insoluble $P = 0.869$, $n = 4–5$; WT insoluble $P = 0.445$, $n = 3–4$) (Figs. 8C and 8D). Similarly, there were no differences in amounts of lumbar P2X7A between WT and SOD1$^{G93A}$ mice at each time point for either fraction, and no significant differences in the amount of lumbar P2X7A over time for SOD1$^{G93A}$ mice (soluble $P = 0.170$, $n = 4–5$; insoluble $P = 0.115$, $n = 4–5$). Conversely, WT mice at 60 and 90 d had significantly less detergent-soluble lumbar P2X7A than at 30 d ($P = 0.007$, $n = 3–4$), and at 90 and 120 d had significantly more detergent-insoluble lumbar P2X7A than at 30 d ($P = 0.027$, $n = 3–4$) (Figs. 8C and 8D).

## DISCUSSION

In the current study, BBG treatment beginning at pre-onset of clinical ALS (62–64 d) delayed weight loss and prolonged survival in female, but not male, SOD1$^{G93A}$ mice.

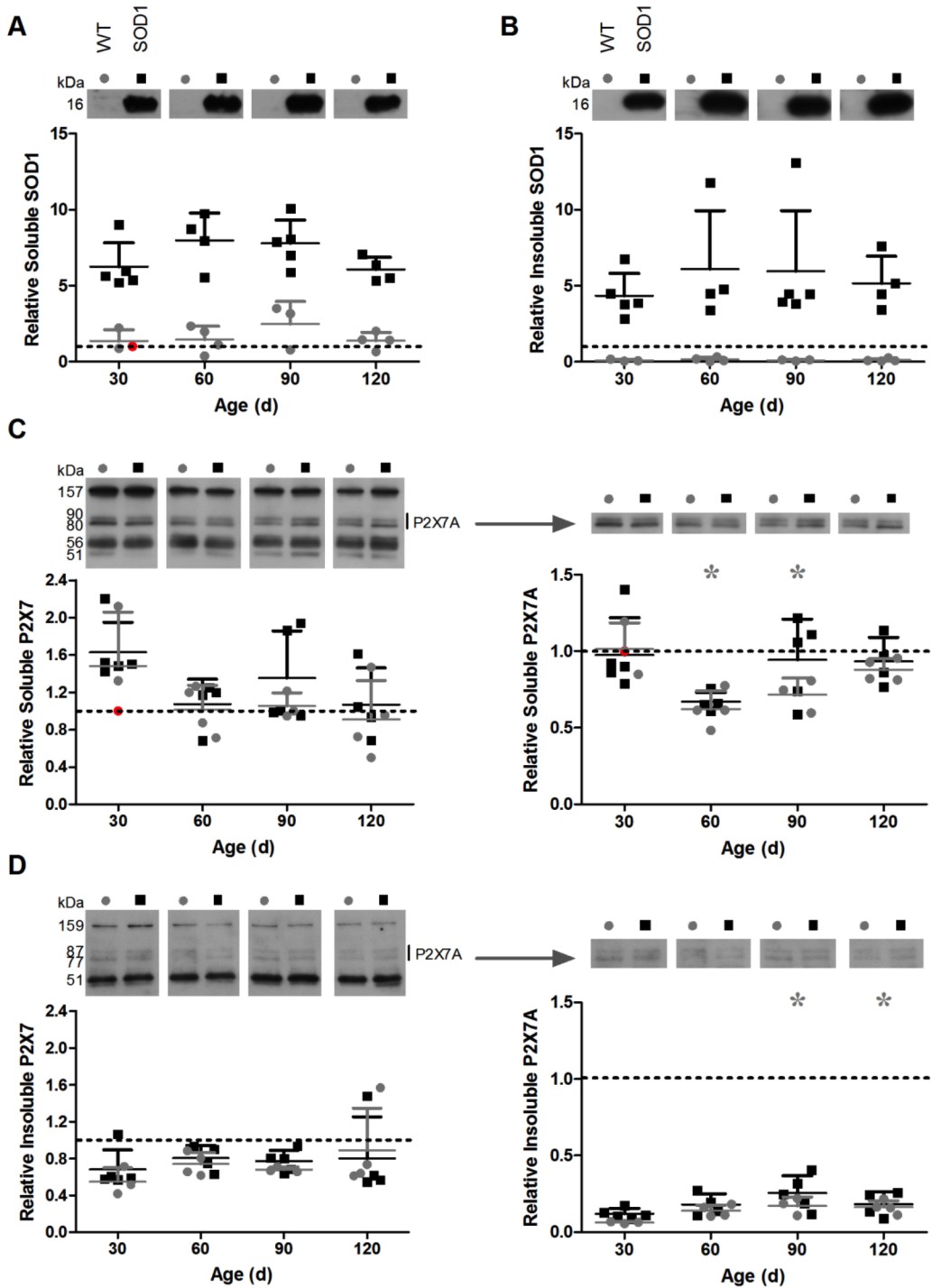

**Figure 8** **The amount of detergent-soluble or -insoluble lumbar SOD1 or P2X7 protein does not change over time in female SOD1$^{G93A}$ mice.** (A, C) Soluble or (B, D) insoluble protein fractions from homogenised female wild type (WT) or SOD1$^{G93A}$ mouse lumbar spinal cords at different ages were separated by SDS-PAGE under reducing conditions, transferred to nitrocellulose membranes, and incubated with (A, B) anti-SOD1 or (C, D) anti-P2X7 pAbs. Representative blots are indicated. Results shown as mean (A, C) soluble or (B, D) insoluble SOD1 or P2X7 ±SD, $n = 3$–5 mice; *$P < 0.05$ compared to corresponding 30 d. Amounts shown are relative to corresponding soluble proteins of one WT mouse at 30 days of age (indicated in red). Symbols represent individual animals.

Treatment had no effect on ALS score or motor coordination in either sex. Furthermore, BBG treatment had no effect on motor neuron loss, microgliosis, lumbar SOD1 or P2X7 protein amounts, splenic leukocyte immunophenotype, or P2X7 expression in the spleen or liver at end-stage. BBG treatment partially, but not significantly, increased serum MCP-1. Together, this suggests a limited efficacy of BBG on ALS progression as used in the current study. Furthermore, there was no difference in the amount of lumbar P2X7 protein observed in SOD1$^{G93A}$ mice of different ages or between SOD1$^{G93A}$ and WT mice at any age.

The therapeutic effects of BBG in SOD1$^{G93A}$ murine models of ALS have been reported in two other independent studies (*Apolloni et al., 2014*; *Cervetto et al., 2013*). In both these studies, treatment with BBG delayed disease onset, improved motor coordination, disease scores and motor neuron survival, and reduced weight loss and microgliosis. Comparisons of these studies with the current study highlight important parameters effecting drug efficacy in murine ALS models, which should be considered in future studies.

One important parameter is gender. In the current study, BBG treatment slowed weight loss and extended survival in female, but not male, mice. This is consistent with the extended lifespan of female heterozygous and homozygous P2X7 knock-out (P2X7$^{KO}$)/SOD1$^{G93A}$ mice compared to P2X7$^{WT}$/SOD1$^{G93A}$ mice (*Apolloni et al., 2013a*). Gender also affected P2X7 antagonism in another study investigating BBG efficacy in the SOD1$^{G93A}$ murine model of ALS. However, in contrast to the current study, BBG significantly delayed the time taken for 10% weight loss and a decline in motor coordination to be observed in male, but not female, mice (*Cervetto et al., 2013*). Furthermore, this gender dependency was not observed for any other reported outcomes, including weight or motor coordination, following BBG treatment in another previous study (*Apolloni et al., 2013a*). Gender-specific outcomes may be impacted by differences in the genetic background of SOD1 mice (*Heiman-Patterson et al., 2005*) or differences in sex hormones such as oestrogens (*Choi et al., 2008*; *Groeneveld et al., 2004*; *Trieu & Uckun, 1999*).

Vehicle composition is another factor that may have influenced the efficacy of BBG observed in the current study compared to previous studies. In the current study, dimethyl sulfoxide (DMSO) was not included in the vehicle for BBG delivery, while DMSO was present in the vehicle solution in both previous studies investigating the therapeutic effects of BBG (*Apolloni et al., 2014*; *Cervetto et al., 2013*). In the current study, DMSO was excluded as BBG was soluble in its absence and given that it is well known to be toxic, especially when used chronically or at concentrations greater than 10% (*Kloverpris et al., 2010*). Due to the fact that intraperitoneally delivered DMSO is able to cross the blood–brain barrier (*Broadwell, Salcman & Kaplan, 1982*), it is possible that the lack of DMSO in the current study prevented adequate BBG blood–brain barrier penetrance. In a study of spinal cord injury, intravenous administration of a similar concentration of BBG in the absence of DMSO resulted in blue colouring in the injury area, but not surrounding tissue (*Peng et al., 2009*). This suggests that BBG entered the site of injury mainly via the disrupted spinal blood–brain barrier in this study, and that DMSO may be required in the drug vehicle for adequate BBG blood–brain barrier penetrance.

The limited efficacy of BBG in the current study may have been due to the dose used. BBG was used at 45.5 mg/kg due to the efficacy of this dose in murine models of other

neurodegenerative conditions, including Huntington's disease and spinal cord injury (*Diaz-Hernandez et al., 2009*; *Peng et al., 2009*). This dose also improved disease outcomes in C57BL/6J × SJL/J hybrid SOD1$^{G93A}$ mice (*Cervetto et al., 2013*). However, in the study of *Apolloni et al. (2014)*, 50 mg/kg BBG had no effect on disease outcomes in C57BL/6J SOD1$^{G93A}$ mice. When the BBG dose was increased to 250 mg/kg in this study, improvements in behavioural scores, motor performance and median disease onset were observed (*Apolloni et al., 2014*). This may indicate that the BBG dose utilised in the current study was insufficient to fully antagonise P2X7 in the CNS of C57BL/6J SOD1$^{G93A}$ mice.

An important consideration when testing therapeutic intervention in ALS mouse models is the timing of treatment commencement. In the current study, BBG treatment was started before the onset of symptoms. In comparison, *Cervetto et al. (2013)* began treatment at disease onset, with a similar BBG dosage, number of treatments per week and delivery route utilised as in the current study. Thus, the differences observed between this and the current study may be explained by the timing of treatment commencement. In the study by *Apolloni et al. (2014)*, BBG treatment was started at late pre-onset, resulting in a wide range of beneficial effects on disease progression. Whilst this study only observed these effects by utilising a five-fold higher dose of BBG than that used in the current study, these high BBG doses were not effective when began during the asymptomatic or pre-onset phases of disease. Collectively, this suggests a very tight window for therapeutic intervention targeting P2X7. This is further supported by the exacerbated disease progression observed in SOD1$^{G93A}$ mice lacking P2X7 (*Apolloni et al., 2013a*), and suggests that P2X7 may play a dual role in ALS progression. In this regard, P2X7 activation may be neuroprotective in the initial stages of disease, but neuroinflammatory or neurotoxic as the disease progresses. This is consistent with the trophic and toxic actions mediated by P2X7 in surveilling and activated microglia, respectively (*Bartlett, Yerbury & Sluyter, 2013*; *Gendron et al., 2003*; *Hide et al., 2000*; *Monif et al., 2009*; *Shieh et al., 2014*).

Microgliosis and loss of motor neurons are two major hallmarks of disease in ALS. Whilst these disease hallmarks were observed in the current study, there were no differences at end-stage between untreated and BBG-treated mice. In contrast, BBG treatment reduced lumbar microglia activation and inflammatory microglial markers, enhanced neurotrophic factors and improved motor neuron survival at end-stage in the study by *Apolloni et al. (2014)*. This is consistent with the anti-inflammatory effects resulting from P2X7 inhibition *in vitro*, whereby BBG treatment reduced the release of pro-inflammatory factors and subsequent toxicity of SOD1$^{G93A}$ microglia towards neuronal cells (*Apolloni et al., 2013b*; *D'Ambrosi et al., 2009*). In the current study, motor neuron and microglial numbers were assessed without stratifying for gender due to the small n values of available samples. However, given the limited clinical efficacy of BBG, increasing the n value to allow for gender stratification may not be worthwhile. Furthermore, there is limited evidence that gender influences motor neuron loss (*Veldink et al., 2003*).

An accumulation of SOD1-containing aggregates is another major hallmark of disease in ALS. In the current study, detergent-insoluble SOD1 was present in lumbar spinal cords of SOD1$^{G93A}$ mice at end-stage. However, there were no differences in the amounts of detergent-insoluble SOD1 between saline- and BBG-treated SOD1$^{G93A}$ mice. Consistent

with this data, ablation of P2X7 does not modify the lumbar SOD1 content in SOD1$^{G93A}$ mice (*Apolloni et al., 2013a*). BBG, but not other P2X7 antagonists, prevents protease-resistant prion protein accumulation in scrapie-infected microglial and neuronal cell lines, and in the brains of prion-infected mice (*Iwamaru et al., 2012*). This suggests that BBG has anti-aggregation properties, independent of its P2X7 interactions. However, given the current data, the anti-aggregation properties of BBG may be restricted to anti-prion activities, consistent with a molecular framework analogous to other anti-prion compounds (*Iwamaru et al., 2012*).

P2X7 protein was detected in the lumbar spinal cord of SOD1$^{G93A}$ mice treated with either saline or BBG. Similar amounts were detected between treatments, consistent with the lack of effect of BBG on lumbar spinal P2X7 amounts in other studies (*Apolloni et al., 2014*). In the current study, a number of bands were detected in addition to the expected full length P2X7A isoform (73–89 kDa). These proteins were approximately 160, 80, 60 and 50 kDa in size, and may correspond to P2X7 dimers, non-glycosylated P2X7, P2X7 splice variants or P2X7 breakdown products. C terminal truncated P2X7 variants have recently been identified in mice (*Masin et al., 2012*). One of these, termed P2X713B, is detected in the mouse CNS and has a molecular mass of approximately 60 kDa (*Masin et al., 2012*), similar to one of the proteins detected in this current and other studies (*Barth et al., 2007*). Interestingly, when expressed in human embryonic kidney (HEK)-293 cells, the majority of P2X713B was retained in the endoplasmic reticulum and not efficiently trafficked to the cell surface (*Masin et al., 2012*). This may explain differences in the amount of 60 kDa protein in detergent-soluble and insoluble fractions in the current study. However, further work would be required to identify whether this band does indeed correspond to a P2X7 splice variant.

In the current study, the amount of lumbar spinal cord P2X7 did not change significantly with age in female SOD1$^{G93A}$ mice. It has been reported that there is no sexual dimorphism in P2X7 expression at any age (*Crain, Nikodemova & Watters, 2009*). However, while microglial P2X7 expression was maintained at similar levels from 21 to 365 d in healthy C57BL/6 mice in one mixed-sex study (*Crain, Nikodemova & Watters, 2009*), another found that P2X7 protein increased with age in healthy Sprague–Dawley rats (*Lai et al., 2013*). Furthermore, similar amounts of lumbar P2X7 protein were present in spinal cords from WT and SOD1$^{G93A}$ mice at all investigated time points in the current study, while P2X7 immunoreactvities have been reported to be increased in spinal cords from (post-mortem) humans with ALS and advanced-stage transgenic SOD1$^{G93A}$ rats (*Casanovas et al., 2008*; *Yiangou et al., 2006*).

In the current study, P2X7 was detected in both detergent-soluble and -insoluble fractions. The presence of detergent-insoluble P2X7 has been reported in mouse lung alveolar epithelial cells, HEK-293 cells transfected with WT P2X7, mouse peritoneal macrophages and rat submandibular glands (*Barth et al., 2007*; *Garcia-Marcos et al., 2006*; *Gonnord et al., 2009*). In these studies, it was suggested that P2X7 partially localised with detergent-resistant membranes, or lipid rafts, at the plasma membrane. This association was reported to be dependent on the post-translational modification of P2X7 by palmitic acid (*Gonnord et al., 2009*). In the current study, the amount of detergent-insoluble 75 kDa P2X7 was relatively low, suggesting that the majority of 75 kDa P2X7 was localised in the plasma

membrane. Determining whether the detergent-insoluble 75 kDa protein represented lipid raft-associated or intracellular P2X7 was beyond the scope of the current work.

BBG treatment increased serum MCP-1 in SOD1$^{G93A}$ mice by 1.4-fold compared to saline treatment; a difference that approached statistical significance ($P = 0.052$). MCP-1 is up-regulated in ALS mice (*Kawaguchi-Niida et al., 2013*) and in patients with ALS (*Henkel et al., 2004*; *Nagata et al., 2007*; *Wilms et al., 2003*), where it is thought to drive disease progression. Whilst *in vitro* studies have showed that pre-incubation with BBG can reduce P2X7-mediated accumulation of intracellular MCP-1 in rat microglia (*Fang et al., 2011*) and prevent P2X7-mediated MCP-1 release from murine microglia (*Shieh et al., 2014*). Thus, the current observation that serum MCP-1 was potentially increased in SOD1$^{G93A}$ mice following BBG treatment was unexpected. Possible explanations for this finding is that BBG treatment *in vivo* affects pathways other than the ATP-P2X7 signalling axis to promote MCP-1 production or alters the circulatory system to extend the half-life of serum MCP-1.

## CONCLUSIONS

In conclusion, BBG treatment reduced body weight loss and prolonged survival in female SOD1$^{G93A}$ mice. This represents the first report of an extension in survival following P2X7 blockade in an ALS mouse model. Given that this result was seen only in females, the importance of considering sex in pre-clinical studies is clear, consistent with reports by others (*Cervetto et al., 2013*). Despite this, in the current study, BBG had no effect on body weight loss or survival in male SOD1$^{G93A}$ mice, or on ALS score, motor coordination, motor neuron loss, microgliosis, lumbar SOD1 or P2X7, serum MCP-1, splenic leukocyte immunophenotype or P2X7 expression at end-stage in either gender. While the other two studies investigating the efficacy of BBG in ALS found more beneficial effects of this compound at molecular and phenotypic levels, these effects did not correlate with an extension in life span in these studies (*Apolloni et al., 2014*; *Cervetto et al., 2013*). Together, the different outcomes obtained from these three studies highlight how drug regime may influence disease outcomes in murine models of ALS.

## ACKNOWLEDGEMENTS

We are grateful to Masoud Yousefi (University of British Columbia, Vancouver, Canada) for assistance with statistical analysis. We are grateful to Natalie Farrawell (University of Wollongong) for assistance with immunostaining. We are grateful to staff of the Illawarra Health and Medical Research Institute (Wollongong, Australia) and Phillip Mullany (University of Wollongong) for technical assistance.

### Funding

This project was supported by the University of Wollongong, Wollongong, Australia. Rachael Bartlett was a recipient of a Global Challenges Scholarship, University of Wollongong. Justin J. Yerbury was supported by the Australian Research Council

(DE120102840). The funders had no role in study design, data collection and analysis, decision to publish, or preparation of the manuscript.

## Grant Disclosures

The following grant information was disclosed by the authors:
University of Wollongong, Wollongong, Australia.
Global Challenges Scholarship, University of Wollongong.
Australian Research Council: DE120102840.

## Competing Interests

The authors declare there are no competing interests.

## Author Contributions

- Rachael Bartlett conceived and designed the experiments, performed the experiments, analyzed the data, wrote the paper, prepared figures and/or tables, reviewed drafts of the paper.
- Vanessa Sluyter performed the experiments, reviewed drafts of the paper.
- Debbie Watson performed the experiments, analyzed the data, contributed reagents/materials/analysis tools, reviewed drafts of the paper.
- Ronald Sluyter conceived and designed the experiments, performed the experiments, analyzed the data, contributed reagents/materials/analysis tools, wrote the paper, reviewed drafts of the paper.
- Justin J. Yerbury conceived and designed the experiments, performed the experiments, analyzed the data, contributed reagents/materials/analysis tools, wrote the paper, prepared figures and/or tables, reviewed drafts of the paper.

## Animal Ethics

The following information was supplied relating to ethical approvals (i.e., approving body and any reference numbers):
Animal Ethics Committee of the University of Wollongong.
Approval numbers AE11/29 and AE12/09.

## Data Availability

The raw data has been supplied as Data S1.

## Supplemental Information

Supplemental information for this article can be found online at http://dx.doi.org/10.7717/peerj.3064#supplemental-information.

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
