# Peer review of "P2X7 antagonism using Brilliant Blue G reduces body weight loss and prolongs survival in female SOD1G93A amyotrophic lateral sclerosis mice"

_PeerJ, doi:10.7717/peerj.3064_

## Round 0.1 · original submission · Major Revisions

Both reviewers have requested additional data analysis, inclusion of new data and further discussion. Reviewer #1 has also requested that a number of experiments are repeated using an additional dose (250 mg/kg) of BBG.

Please see their comments below and include the additional information requested or justify why this is not required.

Reviewer 1 ·

Basic reporting

The author has compared this study to two previous independant studies done by Apolloni et al., 2014 and Cervetto et al., 2013. However the experimental results shown here are unable to justify the main hypothesis of the manuscript which is to identify the role of P2X7 receptors in ALS progression. The author do clearly describes that by using Brilliant Blue G which is a P2X7 antagonist helps in reducing the body weight loss in female SOD1G93A mice but is unable to connect the link through different assays used in this study like immunohistochemistry, microglia density, serum MCP-1 concentrations in SOD1 mice or the P2X7 protein expression as all the results are insignificant.
Clearly according to PeerJ policy, negative or inconclusive data are acceptable but it is kindly suggested to change the angle of the title or discussion because its very hard to believe that Brilliant Blue G is playing its effect through P2X7 antagonism even though previous literature do suggests so or might do some more experiments with a higher dose of BBG to check its effect as done by Appoloni et al., in 2014.

Experimental design

The experiments are very nicely designed throughout this study. The author has put in a lot of effort in this study looking at different changes in protein expression of P2X7 and SOD1, serum MCP1 levels, distribution of splenic leukocytes etc. However the author couldnt see any significant difference between saline and BBG treatment that could convince that the BBG effect seen is actually mediated through P2X7 block and not a random effect of the dye alone. Even the results with weight loss in Figure 1 and 2 are badly interpreted
I have the following suggestions in this regard. The author has only used one concentration of Brilliant Blue G throughout the whole study where as Apolloni et al. (2014) also used a higher dosage of 250 mg/kg. Admitting that the 45 mg/kg of dose was too small to see any significant effects, it should be useful to see the same effects by injecting a higher dose of 250 mg/kg and see if the amount of microgliosis, P2X7 expression and leukocytes expression change.

Validity of the findings

I am happy that PeerJ allows you to look at the raw data. By reanalysing the raw data for Figure 1A and 2A using Prism5 software and by using the same statistical analysis as done by the author (Non paired t-test) I am unable to see any significance in P value score for Figure 1A. The P value score for Figure 1A came out to be 0.14 using two tailed unpaired student t-test however it is mentioned to be 0.014 in the manuscript. Same goes for Figure 2A for female mice where the P value mentioned in the text is 0.006 but after checking the analysis it came out to be 0.03 which is still significant. The author clearly needs to recheck all the analysis and mention clearly what analysis parameters are used.

·

Basic reporting

No comments

Experimental design

Well-performed

Validity of the findings

In order to sustain the lack of effect on microgliosis, the authors should include the analysis of for instance CD68 positive cells in lumbar spinal cord.

Figure 3A: please provide a better image for motor neurons quantification.

Additional comments

Ms. ID: PeerJ 13295
Title: P2X7 antagonism using Brilliant Blue G reduces body weight loss and prolongs survival in female SOD1G93A amyotrophic lateral sclerosis mice
Corresponding Authors: Ronald Sluyter and Justin J Yerbury
The manuscript describes the effects of the P2X7 receptor antagonist Brilliant Blue G (BBG) in the SOD1G93A mouse model of ALS, showing that BBG treatment reduces body weight loss and slightly prolongs survival in female, but not male, SOD1G93A mice, although not improving clinical score or motor coordination in either sex. Moreover the authors demonstrate that BBG treatment has no effect on microgliosis and motor neuron survival at end stage of the disease in SOD1G93A mice, as well as on the amounts of lumbar spinal cord SOD1 and P2X7 and on the serum monocyte chemoattractant protein-1 and splenic leukocyte populations.
The authors have performed a generally well-done experimental study and the paper adds further insights on the complex role of P2X7 receptor in ALS disease, yet investigated in other papers under different experimental conditions, as they report.
However, the authors should discuss why the slight improvement in survival (<10% and only in female mice) is not sustained by any improvement in behavioural scores or in central and neither peripheral hallmarks of ALS disease.
This is of interest in the light of data indicating the emerging role of P2X7 receptor in pathomechanisms involved in ALS disease.

Revisions:
The improvement in female survival should be indicated in the abstract as % value.
How the authors explain the increased amount in MCP-1 in BBG-treated mice?

In order to sustain the lack of effect on microgliosis, the authors should include the analysis of for instance CD68 positive cells in lumbar spinal cord.

Figure 3A: please provide a better image for motor neurons quantification.

---

## Round 0.2 · Minor Revisions

Thank you for making the changes to the manuscript as recommended by your reviewers. Savina is happy with the manuscript but reviewer #1 has requested an additional bar graph in figure 1a and 2a. I hope that you can make these changes.

Reviewer 1 ·

Basic reporting

OK

Experimental design

OK

Validity of the findings

Data is acceptable if following changes are made mentioned below

Additional comments

It would be helpful to include another bar graph for figure 1a and 2a showing mean percentage body weight comparisons to avoid further confusion regarding statistical comparisons for the reader.

·

Basic reporting

No comment

Experimental design

No comment

Validity of the findings

No comment

Additional comments

The authors have addressed all the requested comments.

---

## Round 0.3 · accepted · Accept

Thank you for adding the additional bar graphs suggested by reviewer #1. The manuscript is now ready for publication.